# CoSe-Co: Sentence Conditioned Generative CommonSense Contextualizer for Language Models

**Rachit Bansal**$^{D\dagger}$                                                                RACBANSA@GMAIL.COM

**Milan Aggarwal**$^{A}$                                                              MILAGGAR@ADOBE.COM

**Sumit Bhatia**$^{A}$                                                                SUMBHATI@ADOBE.COM

**Jivat Neet Kaur**$^{M\dagger}$                                                          JIVATNEET@GMAIL.COM

**Balaji Krishnamurthy**$^{A}$                                                        KBALAJI@ADOBE.COM

$^{A}$*Media and Data Science Research Labs, Adobe*

$^{D}$*Delhi Technological University, New Delhi, India*

$^{M}$*Microsoft Research India*

$^{\dagger}$*Work done during an internship at Adobe MDSR*

## Abstract

Pre-trained Language Models (PTLMs) have been shown to perform well on natural language reasoning tasks requiring commonsense. Prior work has leveraged structured commonsense present in knowledge graphs (KGs) to assist PTLMs. Some of these methods use KGs as separate static modules which limits knowledge coverage since KGs are finite, sparse, and noisy. Other methods have attempted to obtain generalized and scalable commonsense by training PTLMs on KGs. Since they are trained on symbolic KG phrases, applying them on natural language text during inference leads to input distribution shift. To this end, we propose a task agnostic sentence-conditioned generative **Co**mmon**Se**nse **Co**ntextualizer (CoSe-Co), which is trained to generate contextually relevant commonsense inferences given a natural language input. We devise a method to create semantically related sentence-commonsense pairs to train CoSe-Co. We observe commonsense inferences generated by CoSe-Co contain novel concepts that are relevant to the entire sentence context. We evaluate CoSe-Co on multi-choice QA and open-ended commonsense reasoning tasks on the CSQA, ARC, QASC, and OBQA datasets. CoSe-Co outperforms state-of-the-art methods in both these settings, while being task-agnostic, and performs especially well in low data regimes showing it is more robust and generalises better.

## 1. Introduction

Given some natural language text, a central aspect of human reasoning is based upon understanding the text to expand certain salient concepts by the virtue of commonsense and logical inference capabilities. This process is often implicit yet plays a large role in routine interactions and answering pragmatic questions correctly. A body of prior work [Bosselut et al., 2019, 2021, Wang et al., 2020] has focused on imbibing such inference capabilities into popular language models (LMs) [Devlin et al., 2019, Radford et al., 2018] by leveraging commonsense knowledge sources. This enables the LMs to become commonsense aware in addition to possessing textual knowledge [Jiang et al., 2020, Petroni et al., 2019, Roberts et al., 2020] and semantic understanding [Li et al., 2021].

There have been various attempts to leverage structured knowledge present in commonsense knowledge graphs (KGs) such as ConceptNet [Speer et al., 2017]. Such works have

primarily focused on either of the following two aspects - (i) learning a model to generate knowledge on-demand, or (ii) solving a particular downstream task by retrieving commonsense knowledge from a KG. Generative methods like COMET [Bosselut et al., 2019] leverage the expressiveness of LMs to learn commonsense knowledge through training on symbolic concepts in a KG and the relations between them. However, such methods are often used to perform inference on sentences [Bosselut et al., 2021] which results in train-inference input distribution shift. On the other hand, task-specific methods like QA-GNN [Yasunaga et al., 2021] uses an architecture which relies heavily on the structure of a downstream task like question-answering to leverage the static knowledge in a KG. In a nutshell, the former has the ability to generate novel concepts on-demand but their applicability to a downstream task is not straightforward; the latter is designed for a particular setting and does not generalize well beyond a specific task.

To address these limitations, we propose a CommonSense Contextualizer- **CoSe-Co**, a generic generative framework which outputs commonsense knowledge given natural language sentence(s) as input context. The generated commonsense knowledge is in the form of paths i.e., sequence of entities connected through relations. The relation vocabulary is determined based on the KG schema used for training CoSe-Co. Since CoSe-Co outputs commonsense inferences conditioned on a sentence as input, it has the capability to dynamically select entities/phrases from the input sentence as well as generate novel unseen entities relevant to the sentence while conditioning them to the path being generated. To achieve this, we first create sentence-path pairs by- 1) sampling paths from an underlying KG; 2) sampling a subset of entities from the path; 3) retrieving & filtering sentences from Wikipedia that contain the sampled entities and are semantically similar to the path; and finally leverage these pairs to train a pre-trained language model to generate paths given a sentence as input. The trained CoSe-Co is task agnostic and can be used to generate on-demand commonsense knowledge from a given text for any downstream application.

The reasoning ability and generalizability of LMs possessing commonsense have been commonly analysed on the task of question-answering (QA) [Sap et al., 2019b, Talmor et al., 2019, Mihaylov et al., 2018]. In order to compare our model in-line with prior work, we evaluate our framework on two diverse QA benchmarks. The first is multi-choice QA on the CSQA dataset [Talmor et al., 2019] where given a question and a set of answer choices, the task is to select the correct choice. However, in this setting, often more than one choice is a suitable answer. To mitigate this, open-ended commonsense reasoning (OpenCSR) [Lin et al., 2021] was introduced where every question is annotated with a set of possible answers. Hence as our second QA setting, we compare our framework with baselines for OpenCSR on ARC [Clark et al., 2018], QASC [Khot et al., 2020] and OBQA [Mihaylov et al., 2018] datasets. Although CoSe-Co can be used across a wide variety of tasks such as machine translation, dialogue systems, etc., we leave this exploration as future work. Our contributions can be summarised as:

1. We propose a task agnostic sentence conditioned generative framework (**CoSe-Co**) which outputs commonsense inferences corresponding to natural language input. We also provide a method to create paired sentence-path data for training CoSe-Co.

2. CoSe-Co leverages the semantic understanding present in an LM to produce novel entities and concepts as commonsense knowledge since it is not strictly tied to the entities in the KG used for training it owing to its generative nature.

3. Our framework adapts to text input in downstream tasks since it does not suffer from train-inference input distribution shift which other generative commonsense LMs face.

4. We outperform prior approaches for multi-choice QA and OpenCSR tasks. Specifically for CSQA, we show that our method performs even better than baselines in low training data regimes demonstrating it is robust and able to generalize better.

## 2. Related Work

Commonsense Knowledge Graphs (CSKG) like ConceptNet [Speer et al., 2017] and ATOMIC [Sap et al., 2019a] are structured knowledge sources comprising of entities in the form of symbolic natural language phrases, connected through a set of relations. This structured knowledge is usually leveraged to perform structured explainable reasoning [Ren* et al., 2020, Ren and Leskovec, 2020] and provide additional context in NLP tasks [Bao et al., 2016, Sun et al., 2018, Lin et al., 2019]. We focus on ConceptNet in this paper and plan to subsequently extend our methodology to other CSKGs. Recently, many Pre-Trained Language Models (PTLMs) have shown remarkable success in tasks requiring commonsense [Zellers et al., 2018, Bhagavatula et al., 2019, Huang et al., 2019]. However it has remained unclear as to whether this performance is due to reasoning abilities or if it is because of capturing correlation in data [Zhou et al., 2020, Mitra et al., 2019, Niven and Kao, 2019].

To address the above issues, LM + KG systems have been extensively explored [Feng et al., 2020, Wang et al., 2018, Lv et al., 2020] to combine informative knowledge from KG with semantic understanding of LM. However, using a static KG with an LM faces several shortcomings such as the KG might be noisy, sparse and may not be exhaustive enough to contain structured knowledge about all facts. Bosselut et al. [2019] took the initial strides to address this by leveraging the unstructured knowledge contained in an LM to generalise the knowledge contained in a KG. Their proposed model, COMET, is trained to generate tail entity given head entity and relation in a KG triple. This way it learns to generate novel entity nodes starting from a seed node during inference. However, it is not straightforward how such a model could be extended to a downstream task. Follow-up work devised an approach for context-driven QnA [Bosselut et al., 2021], where COMET is used to generate commonsense inferences, but there is a big mismatch between the training and inference input distributions since COMET is trained on symbolic text (KG triples) while inferred on natural language sentences. Additionally, having being trained on single triples, it is likely to face challenges in scenarios where multi-hop reasoning is required.

Most recently, Yasunaga et al. [2021] proposed a framework to leverage a given static CSKG for QA. Although, such an approach avoids distillation of knowledge and makes it easier to extract evidence from a CSKG on-demand, the knowledge source is static which undesirably restricts the knowledge that can be garnered.

Closest to our approach is the method proposed by Wang et al. [2020], aimed at generating commonsense inferences as 'paths' between source and target entities. Designed specifically for question-answering, their path generator is trained to connect pairwise ques-

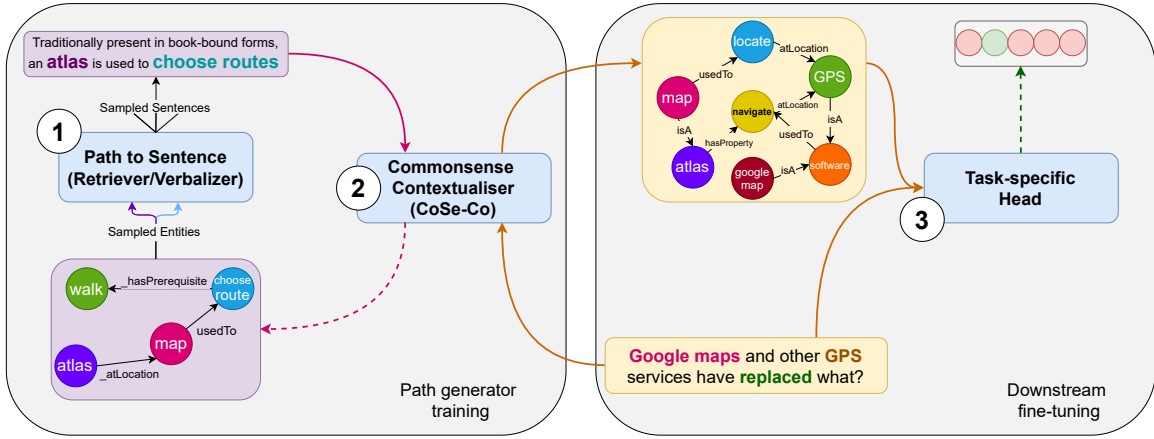

Figure 1: Our proposed approach consists of three main modules: (i) **Path to Sentence Retriever**, to create the training data for CoSe-Co, (ii) **Co**mmon**Se**nse **Co**ntextualizer (CoSe-Co), to generate commonsense inferences from a natural language sentence, and (iii) **Task-specific Head**, to leverage generated knowledge as additional context in a downstream task.

tion and answer choice entities through commonsense axioms just like paths in a KG. Much like COMET, novel knowledge entities are observed in the generated inferences, while being multi-hop in nature. However, being conditioned only on singleton question-choice entity pairs at a time, the generated paths do not capture the overall context of the question. Two distinct questions, 'Google maps and other GPS services *have replaced what*?' and 'Google maps and other GPS services *are useful for*?', would produce very similar paths for the same answer choices which is highly undesirable and not the case with CoSe-Co.

## 3. Methodology: CoSe-Co Framework

We design a **Co**mmon**Se**nse **Co**ntextualizer (CoSe-Co) to generate commonsense inferences given a sentence as input. Commonsense KGs such as ConceptNet [Speer et al., 2017] comprise of symbolic knowledge in the form of concepts and entities represented as nodes linked through directed relational edges. However, there is no dataset that maps a sentence to a corresponding commonsense-relevant sub-portion of an underlying KG. To bridge this gap, we first devise a methodology to create a dataset comprising of contextually related sentence-path pairs. This dataset is then used to train a LM based commonsense knowledge model which learns to map the sentence to a commonsense inference path $p = \{e_1, r_1, e_2, r_2, ..., e_n, r_n\}$ such that $e_i \in E$ and $r_i \in R$ where $R$ is the set of relations in the KG. Here, $E = E_{KG} \cup E_{novel}$ where $E_{KG}$ represents the entities in KG and $E_{novel}$ are entirely novel concepts which includes concepts present in input sentence but not in the KG. CoSe-Co learns to generate such novel concepts owing to our designed mechanism and also since it is based on a pre-trained LM possessing a larger vocabulary as compared to the KG. During inference, given a sentence $s$ (which can be a question, dialogue prompt or normal sentence), CoSe-Co generates a set of commonsense inferences $I = \{i_1, i_2, ..., i_n\}$ which can

be augmented as additional context with $s$ to perform downstream tasks. An overview of our framework is shown in Figure 1. We now explain each step in detail.

## 3.1 Sentence-Path Paired Dataset Creation

In order to train CoSe-Co, we create the paired dataset by performing a reverse mapping from structured knowledge paths in the KG to sentences. Specifically, we start by performing a random walk in ConceptNet to extract a set of multi-hop paths, $P = \{p_1, p_2, \cdots, p_n\}$, where length of each path is constrained in the range $[k_1, k_2]$, where $k_1 = 2$ and $k_2 = 5$. To avoid paths that do not convey useful information and filter noise, we employ relational heuristics like in Wang et al. [2020] such as removing generic relations etc., which limits the size of $P$ to $28M$. Separately, we extract sentences from Wikipedia articles ($\sim 5M$), to get a set of $92.6M$ sentences. These sentences are then indexed using Solr[1] which is then used for retrieving sentences given a path. We refer to this indexed corpus as $S \in \{s_1, s_2, \cdots, s_n\}$.

We map each path in $P$ to a subset of sentences in $S$, such that sentence-path pairs obtained are used to train CoSe-Co. Consider a path $p_i$ containing $m$ entities $(e_1, \cdots, e_m)$ and $m-1$ relations $(r_1, \cdots, r_{m-1})$. While retrieving relevant sentences, we extract multiple queries from $p_i$ according to two types of query templates - $Q1$ and $Q2$. In $Q1$, we extract non-contiguous entity-relation triples from $p_i$ of the form $\{(e_i, r_i, e_{i+2})\}$ and $\{(e_i, r_{i+1}, e_{i+2})\}$. This is to ensure relation information is also captured while retrieving a sentence. One thing to note is that we do not consider entire path while retrieving sentence in order to ensure better coverage since it is very unlikely that a sentence containing all entities and relations present in a path exists. In $Q2$, we extract only connected entity pairs of the form $\{(e_i, e_{i+1})\}$. For each query $q$ obtained from $p_i$ according to $Q1$ and $Q2$, we query Solr and select sentences containing entities present in $q$ as first step. Subsequently, we rank retrieved sentences based on similarity between their embedding and the embedded representation of the corresponding query $q$ using SBERT [Reimers and Gurevych, 2019]. We select maximum of top 10 sentences corresponding to each path and obtain a final set of $\sim 200K$ path-sentence pairs. Our query template based sentence retrieval enforces CoSe-Co to both extrapolate and interpolate concepts while generating path given a paired sentence as input.

## 3.2 Sentence → Commonsense Path Generator

We use a pre-trained language model to initialise the path generator to leverage the textual knowledge and semantic understanding [Petroni et al., 2019, Jiang et al., 2020, Roberts et al., 2020] they posses. Specifically, we fine-tune the T5-base language model [Raffel et al., 2019] owing to its inherently generative nature. As shown in Figure 2a, we append the prefix '*convert sentence to path:*' before each sentence query to T5 and train it to generate the corresponding path. More formally, given a sentence $s$ prepended with the prompt as input to T5 encoder, it is fine-tuned to generate a path token $p_t$ at decoder time-step $t$ by jointly conditioning over encoder outputs and past tokens in the path $p_{<t}$ through the usual cross-entropy loss. We also design a variant where we randomly select an entity that co-occurs in the input sentence and the target paths, and mask it in the input sentence. This masking is performed only for a certain number of sentences in the dataset and is controlled by a

---

1. https://solr.apache.org/

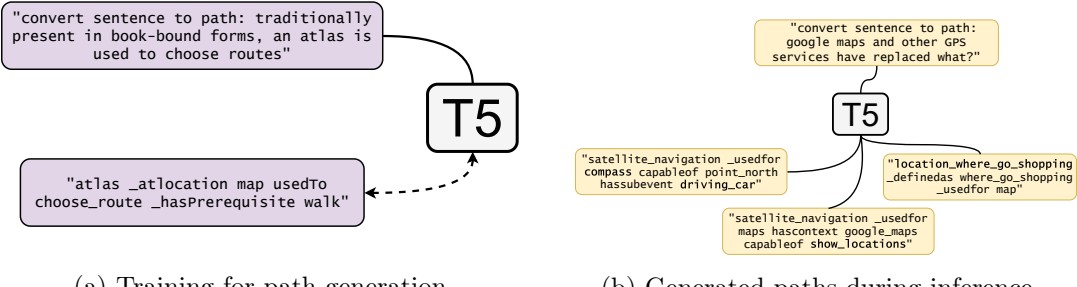

(a) Training for path generation     (b) Generated paths during inference

Figure 2: Adapting T5 as **CoSe-Co** to generate commonsense inferences.

hyper-parameter, $p_{mask}$. The model is thus trained to generate the path that also contains the masked entity. The goal behind masking is to induce an inductive bias in CoSe-Co to identify the masked entity while relating it with other concepts during path generation[2].

### 3.3 Path Decoding During Inference

As in most sequence generation tasks, teacher forcing is used to train the model, while a decoding strategy is used to generate diverse outputs during inference. To maximise the additional contextual knowledge obtained from paths for each sentence during a downstream task, we generate and make use of multiple paths. In order to maximise the diversity among paths while not losing out on relevance, we implement a path-specific variant of beam search, *diverse-path search*. Diversity is ensured in diverse-path search by sampling top-$k$ most probable tokens at the first level itself and then picking the most probable path forward for each one of them, thus returning $k$ paths. This approach is motivated by the fact that when sampling entities for a path, the initial entities influence the overall semantics of the path much more significantly than intermediate entities.

## 4. Experiments and Evaluation

### 4.1 Implementation Details

CoSe-Co is built upon T5-base [Raffel et al., 2019], which is a transformer-based generative model pre-trained on the Colossal Clean Crawled Corpus (C4; Raffel et al. [2019]). CoSe-Co consists of 12 encoder-layers, each with 12 attention heads, a hidden-state size of 768, and feed-forward size of 3072– resulting in 220M parameters. The model is fine-tuned on the task of generating paths from sentences until the validation loss across an epoch does not increase, with the maximum number of epochs being 5. The number of paths per sentence, $k$, is set to 5 and $p_{mask}$ to 0.33 based on tuning on the dev set of CSQA. The AdamW optimizer [Loshchilov and Hutter, 2017] is used to train parameters with a learning rate of $5e-4$, weight decay of 0.01 and epsilon of $1e-8$. CoSe-Co is trained on a single A-100 GPU with a batch-size of 8 with 4 gradient accumulation steps. Wikipedia is used to extract sentences and ConceptNet to extract paths to create paired data for training CoSe-Co.

---

2. We perform ablations to compare these variants quantitatively and qualitatively in the Appendix.

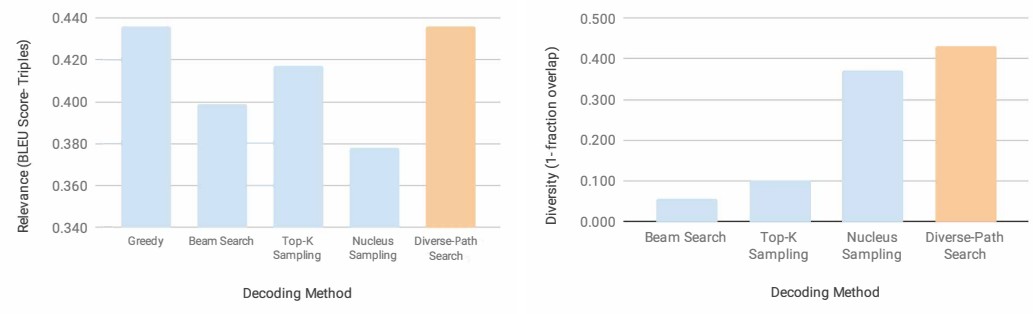

(a) Relevance- BLEU score of generated paths against ground truth paths

(b) Diversity- fraction overlap between top-5 sampled paths.

Figure 3: Evaluation and comparison of generated paths across different decoding strategies

## 4.2 Analysing Generated Paths

We analyse the quality of generated paths on the basis of two metrics - **Relevance** and **Diversity**, evaluated on the test split of paired sentence-path dataset.

- **Relevance** The paths generated using CoSe-Co are compared with the ground truth paths on an unseen split of the dataset. The evaluation is done using BLEU score on triples, i.e., each triple in the path is considered as one uni-gram token.

- **Diversity** Top-$k = 5$ paths are extracted and the amount of fractional overlap is measured between them. The compliment of this term is reported as diversity.

Figure 3 shows the corresponding results. It is observed that paths generated using nucleus sampling are diverse but lack diversity, while an opposite trend is observed for top-k sampling. *Diverse-path search* provides the best balance between relevance and diversity. Table 1 shows qualitative examples of paths generated for few question samples. As can be seen, CoSe-Co generates paths contextually relevant to the question in addition to inferring novel concepts not in ConceptNet.

Commonsense inferences generated by CoSe-Co can be used to augment context in downstream tasks such Question Answering (QA). We evaluate and compare CoSe-Co on multi-choice QA and OpenCSR tasks in the subsections to follow.

## 4.3 CSQA: Commonsense Question-Answering

Following Wang et al. [2020], we use their framework on the CSQA dataset [Talmor et al., 2019] that leverages both structured (commonsense inferences/paths) and unstructured (question and choices) knowledge. Given a question $q$ with corresponding candidate answer choices set $C = \{c_1, \ldots, c_n\}$, we generate commonsense inferences for each choice $c_i$ by augmenting the choice in $q$ and giving the resultant query as an input to CoSe-Co. Specifically, we replace the interrogative element in $q$ with $c_i$. For instance, given the question 'Google maps and other GPS services have replaced what?', we obtain the inferences corresponding to an answer choice 'atlas', by augmenting the question as: 'Google maps and other GPS services have replaced *atlas*.' and provide this as an input to CoSe-Co. We

| Input | CoSe-Co Outputs |
|---|---|
| What do people aim to do at work? | (work_at_home _capableof people desires work _hasprerequisite **earning_living** causes get_paid _by_job)
(get_money_from _capableof people desires work _hassubevent enjoying_day )
(have_to_work _capableof people desires work causes getting_job hasprerequisite payrolls_and_**paying_bills**) |
| What do people typically do while playing guitar? | (playing_guitar causes singing usedfor people capableof feeling_sad)
(playing_guitar hassubevent sing _causesdesire singing _occupation musician genre **folk_rock**)
(play_guitar _usedfor guitar atlocation symphony_halls _or_musical_instruments_or_bands _atlocation people ) |
| Where are you likely to find a hamburger? | (burger _isa hamburger atlocation fast_food_restaurant usedfor eating_food)
(burger_king _usedfor hamburger atlocation fast_food_restaurant isa place capableof **take_car_for_drive**)
(fast_food_restaurant _isa taco_bell product hamburger madeof **wheat_flour_and_salt**) |
| In what Spanish speaking North American country can you get a great cup of coffee? | (bretagne partof north_america _atlocation cup_of_coffee hascontext usa isa country)
(**hot_beverage** _isa coffee atlocation cup_of_coffee hascontext north_america _partof grenada )
(good_coffee hasa **caffiene_in_milk_and_sugar** atlocation in_canada ) |

Table 1: Examples of commonsense inferences obtained from CoSe-Co across questions from the *CommonsenseQA* [Talmor et al., 2019] dataset. Potential answers which are observed in the path itself are highlighted, while context-enriching concepts are **coloured**.

infer $k$ paths corresponding to $c_i$ to get a resultant set of paths - $P_{q-c_i}$. Further, an average over the hidden representations from the final layer of CoSe-Co's decoder are used as path embeddings - $H_S \in R^{k \times h_{T5}}$ to represent the paths in $P_{q-c_i}$.

To embed the question and an answer choice, we use a contextual encoder $E$, such as RoBERTa [Liu et al., 2019] or ALBERT [Lan et al., 2019] to embed the query - '[CLS] $q$ [SEP] $c_i$' corresponding to $c_i$. We extract the [CLS] representations from final hidden layer as $h_{US} \in R^{h_E}$. In order to leverage relevant knowledge from the generated commonsense inferences, the question and choice embedding is used to attend over generated paths as:

$$\alpha_p = Softmax(tanh(H_S W^A) h_{US})$$

$$h_{S'} = \sum_{h \in H_S} \alpha_p^h \cdot h$$

where, $W^A \in R^{h_{T5} \times h_E}$, $\alpha_p \in R^k$ and $h_{S'} \in R^{h_{T5}}$. Finally, a linear layer is applied over the concatenation of $\{h_{US}, h_{S'}\}$ to project it as a scalar. A softmax is taken over concatenation of scalars obtained corresponding to each answer choice to obtain their likelihood followed by cross entropy loss for training.

We use RoBERTa-base [Liu et al., 2019] as the contextual encoder $E$ for most of our experiments. Table 2 shows the results on CSQA. When using the entire training data, we observe that CoSe-Co performs better than all the baselines [3] on the test split. PGQA [Wang et al., 2020] is the most similar to our overall approach since we adapt their framework for CSQA and primarily differ in how the commonsense paths are obtained. We outperform PGQA with a gain of 1.68% in accuracy on the test split signifying the relevance and applicability of inferences generated by CoSe-Co. We perform marginally better than QA-GNN [Yasunaga et al., 2021] (uses static KG as knowledge source) on the test split. However, when compared in low training data regimes, CoSe-Co performs much better

---

3. A portion of the baseline numbers were taken from Wang et al. [2020] and Yasunaga et al. [2021], while the rest were reproduced using their official implementations.

| Methods | 20% Train IHtest (%) | 60% Train IHtest (%) | 100% Train IHdev (%) | 100% Train IHtest (%) |
|---|---|---|---|---|
| RoBERTa-large (w/o KG) | 46.25 (±0.63) | 52.30 (±0.16) | 73.07 (±0.45) | 68.69 (±0.56) |
| + RGCN [Schlichtkrull et al., 2018] | 45.12 (±0.69) | 54.71 (±0.37) | 72.69 (±0.19) | 68.41 (±0.66) |
| + GconAttn [Wang et al., 2018] | 47.95 (±0.11) | 54.96 (±0.69) | 72.61( ±0.39) | 68.59 (±0.96) |
| + KagNet [Lin et al., 2019] | – | – | 73.47 (±0.22) | 69.01 (±0.76) |
| + RN [Santoro et al., 2017] | 45.12 (±0.69) | 54.23 (±0.28) | 74.57 (±0.91) | 69.08 (±0.21) |
| + MHGRN [Feng et al., 2020] | – | – | 74.45 (±0.10) | 71.11 (±0.81) |
| + PGQA [Wang et al., 2020] | 58.25 (±0.43) | 69.66 (±0.97) | 77.53 (±0.47) | 71.19 (±0.49) |
| + QA-GNN [Yasunaga et al., 2021] | 59.08 (±1.25) | 68.70 (±0.62) | 75.54 (±0.42) | 72.29 (±0.43) |
| + CoSe-Co (**Ours**) | **61.20** (±0.19) | **70.23** (±0.40) | **78.15** (±0.23) | **72.87** (±0.31) |

Table 2: Performance comparison on in-house dev (**IHdev**) and test (**IHtest**) split of ***CommonsenseQA*** dataset [Lin et al., 2019]. We report accuracy averaged across 5 runs. The second-best number for each column is underlined while the best number is in bold. We also evaluate by varying the amount of data used while training as depicted in first row.

than QA-GNN with performance gains of $\sim 2\%$ (and $\sim 3\%$ over PGQA) showing that while QA-GNN is more sensitive towards amount of training data used, CoSe-Co is more robust and generalizes better.

## 4.4 OpenCSR: Open-Ended CommonSense Reasoning

For CSQA, often it is the case that multiple choices are suitable answers. However, the model has to select one and gets penalised unfairly if it does not match with the ground truth answer choice. To mitigate this, Lin et al. [2019] re-configured three popular multi-choice question-answering datasets to use them for OpenCSR. The reformulation resembles a cloze task, where interrogative elements are replaced with blanks ("_ _ _") and a set of singleton tokens is labeled as ground truth.

We design a supervised methodology that utilises paths generated by CoSe-Co. The framework is built using a pre-trained generative language model: T5-base [Raffel et al., 2019]. To adapt to the OpenCSR task, we use the masking variant of CoSe-Co to predict the set of paths $P$. For inferring CoSe-Co, the input $q$ is prepared by replacing the blank ("_ _ _") in question with the mask token. The generated paths in $P$ are concatenated with $q$, along with the prefix '*fill mask to answer question:* '. This resultant sentence acts as the input for OpenCSR specific T5-base, which is trained to generate the right answer. During inference, the top-$K$ concepts, determined on the basis of likelihood from decoder's classification layer, are taken as answer candidates predicted by the model.

Table 3 shows the comparison between DrFact[4] [Lin et al., 2021] (current state-of-the-art for OpenCSR) and our supervised framework (as described above) which uses commonsense paths generated by CoSe-Co. More specifically, we evaluate two types of commonsense inferences - 1) 'Paths from CoSe-Co' where the generated paths are concatenated as is and 2) 'Concepts from CoSe-Co' where the relations in the generated paths are filtered and only entities are retained. Since our supervised framework is based on pre-trained T5

---

4. The code for running baseline experiments was taken from their official GitHub repository.

|  | ARC | | | QASC | | | OBQA | | |
|---|---|---|---|---|---|---|---|---|---|
| **Hits@K** | H@10 | H@30 | H@50 | H@10 | H@30 | H@50 | H@10 | H@30 | H@50 |
| DrFact [Lin et al., 2021] | 36.09 | 53.25 | 64.50 | 21.78 | 37.62 | 51.49 | 12.08 | 23.77 | 35.13 |
| T5-base [Raffel et al., 2019] | 49.70 | **67.46** | 71.01 | 33.66 | 47.52 | 53.47 | 17.42 | 29.55 | 37.88 |
| + Paths from CoSe-Co | **50.89** | 63.91 | 69.23 | 30.69 | 47.52 | 56.44 | 20.45 | 34.09 | **45.45** |
| + Concepts from CoSe-Co | 44.97 | 66.86 | **73.37** | **35.64** | 47.52 | **57.43** | **21.21** | **35.61** | 42.42 |
| **Recall@K** | R@10 | R@30 | R@50 | R@10 | R@30 | R@50 | R@10 | R@30 | R@50 |
| DrFact [Lin et al., 2021] | 12.60 | 21.05 | 27.27 | 12.38 | 22.28 | 29.70 | 6.12 | 11.85 | 16.51 |
| T5-base [Raffel et al., 2019] | 15.98 | 28.30 | 33.93 | 18.98 | 26.40 | 30.53 | 8.52 | 14.61 | 18.71 |
| + Paths from CoSe-Co | **16.87** | 27.45 | 33.73 | 17.49 | 28.05 | **33.33** | 9.90 | 16.53 | **22.42** |
| + Concepts from CoSe-Co | 15.12 | **28.99** | **35.21** | **19.64** | 28.05 | 33.00 | **9.96** | **17.35** | 21.10 |

Table 3: Performance comparison on the basis of Hits@K and Recall@K for OpenCSR task [Lin et al., 2021] on three datasets - ARC, QASC and OBQA.

LM, for fair comparison and to probe if performance changes are due to the base LM, we also compare against another baseline where pre-trained T5-base is fine tuned for OpenCSR task. We estimate two metrics as used in Lin et al. [2021] - 1) **Hits@K**: A given sample is considered as correct if at least one concept in the ground truth set matches with at least one concept in predictions set; 2) **Recall@K**: Determined on the basis of how many concepts predicted by a model matches at least one concept in the ground truth set of concepts. We vary the value of K to be 10, 30, 50. We evaluate our method on three datasets[5] - ARC [Clark et al., 2018], QASC [Khot et al., 2020] and OBQA [Mihaylov et al., 2018].

As can be seen, CoSe-Co outperforms both baselines significantly on all three datasets uniformly for both the metrics and different values of K. More specifically, 'Concepts from CoSe-Co' performs better in general which shows concepts generated by CoSe-Co are useful and more relevant for OpenCSR task. Our approach provides performance gains of upto 8%, 6%, 10% in Hits@50 and 8%, 3%, 6% in Recall@50 over DrFact on ARC, QASC and OBQA respectively. Further, even though T5-base baseline performs better than DrFact, the generated commonsense from CoSe-Co augmented with T5 achieves new state of the art on this task with performance gains of upto 2.3%, 3.9%, 7.5% in Hits@50 and 1.2%, 2.5%, 3.7% in Recall@50 over T5-base on ARC, QASC and OBQA respectively.

## 5. Conclusion and Future Work

In this work, we propose a sentence conditioned generative commonsense language model CoSe-Co which outputs knowledge inferences contextually relevant to the sentence given as input. We devise a novel methodology to create paired dataset comprising of semantically related sentence-commonsense path pairs using Wikipedia and ConceptNet to train CoSe-Co. We evaluate on multiple choice commonsense QA (CSQA) and open ended commonsense reasoning tasks performing better than SOTA methods. Specifically, CoSe-Co performs better than baselines in low training data regimes indicating its robustness and generalisability. As future work, we aim to evaluate CoSe-Co on other NLP tasks where commonsense is helpful and also explore & extend our framework to other types of commonsense KGs such as ATOMIC.

---

5. The authors informed that they are still in the process of setting up a leaderboard on the test set. As the official test is hidden, we report numbers on the validation split.

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

# Appendix A. Entity masking while training CoSe-Co

| Input | CoSe-Co Variant | CoSe-Co Outputs |
|---|---|---|
| Google maps and other GPS services have replaced what? | $p_{mask} = 0.0$ | (orienteering _usedfor map _isa **navigation** _hascontext GPS _product gsrc)
(satellite_gear hascontext maps capableof **help_find_places** _usedfor map_store_in_city)
(similar_to_gps _hasproperty google hascontext maps capableof **show_live_places**)
(geospatial _isa navigation _usedfor google hascontext maps capableof show_people_places) |
| Google maps and other GPS services have replaced what? | $p_{mask} = 1.0$ | (gps hascontext **navigation** _usedfor satellite atlocation orbital_stations_in_space)
(satellite_navigation _usedfor ==compass== capableof **point_north** _hassubevent **driving_car**)
(navigation _usedfor satellite _isa gps hascontext navigation _entails competition)
(**location_where_go**_gallery _definedas navigation _usedfor satellite _hascontext gps) |
| Google maps and other GPS services have replaced what? | $p_{mask} = 0.5$ | (==compass== _isa google_maps usedfor **navigation** _hascontext gps isa navigating_map)
(**location_where_go_camping** _isa location _usedfor map _product google_maps)
(satellite_gear _isa GPS usedfor navigation _hascontext gps isa navigating_map)
(navigation_maps _usedfor map _isa navigation_map hascontext navigation) |
| Google maps and other GPS services have replaced what? | $p_{mask} = 0.33$ | (orienteering _usedfor maps _isa google_maps _hasprerequisite looking_through_telescope)
(**location_where_go_shopping** _definedas where_go_shopping _usedfor map)
(navigation_maps _isa maps _usedfor satellite locatednear planet)
(satellite_navigation _usedfor maps _hascontext google_maps capableof show_locations) |
| Google maps and other GPS services have replaced [ ==MASK== ]. | $p_{mask} = 0.33$ | (gps hascontext maps _usedfor satellite locatednear planet)
(navigation_maps isa navigation _usedfor compass capableof **point_north_handle**)
(satellite_navigation _usedfor ==compass== capableof point_north_or_south_hemispheres)
(location_where_go_if_near_beach _definedas map usedfor navigation _mannerof sport) |
| Google maps and other GPS services have [ ==MASK== ] what? | $p_{mask} = 0.33$ | (orienteering _usedfor map _isa google_maps _hascontext gps)
(**location_where_go_if_need_to** _definedas location _isa map usedfor **information**)
(located_in_latin_america _receivesaction israel _language latin_america)
(navigation_maps usedfor find_place _hasprerequisite go_to_market)
(satellite_navigation _usedfor maps capableof **show_locations_and_routes**) |
| [ ==MASK== ] and other GPS services have replaced what? | $p_{mask} = 0.33$ | (navigation_system _isa GPS hascontext astronomy _field edmond_halley)
(location_where_go_if_in_accident _usedfor map _atlocation GPS_systems)
(**radio_frequency_messaging** _isa GPS hasproperty useful)
(receiver partof radio _isa gps hascontext navigation _usedfor ==compass==) |

Table 4: Examples of commonsense inferences obtained for different input forms of the same question from CoSe-Co when trained with different values of $p_{mask}$. Potential answers which are observed in a path are ==highlighted== , while context-enriching concepts are **coloured**.

As described in Section 3.2, the hyper-parameter $p_{mask}$ is used while training CoSe-Co and denotes the probability of masking an entity in the input sentence that originally co-occurs in the target path. This was aimed at developing an inductive bias in CoSe-Co such that it implicitly attempts to predict the masked entity and include it in the generated paths. However, it is vital to tune $p_{mask}$ in order to ensure that this tendency does not overpower our original purpose of generating context-relevant paths. Thus, we perform both qualitative and quantitative analysis of CoSe-Co across different values of $p_{mask}$ as shown in Tables 4 and 5.

Table 4 shows the various kinds of paths obtained from CoSe-Co when trained with different values of $p_{mask}$, across the same original question. A number of observations can be made. First, the paths obtained from the variant which is trained without any masking ($p_{mask} = 0.0$) produces inferences that enrich the overall context of certain entities in question but do not necessarily capture the inter-relation between them and thus the overall intention of the question. With the configurations that are trained with $p_{mask}! = 0$, the various paths capture the overall context in an answer-oriented manner. These configurations also allow us to mask concepts in the original question such that CoSe-Co can exploit the

unmasked entities to direct its generated paths in a manner that best suit the blank. This is evident from the second half of Table 4. When the interrogative element is masked in the first example, the paths are directed towards actually finding the best answer, while when 'Google maps' is replaced in the third example, the paths are clearly focused on predicting concepts related to GPS systems.

| CoSe-Co Variant | CSQA Dev (%) |
|---|---|
| $p_{mask} = 0.0$ (no masking) | 77.52 ($\pm$0.44) |
| $p_{mask} = 1.0$ (masking for all examples) | 77.38 ($\pm$1.17) |
| $p_{mask} = 0.67$ | 77.61 ($\pm$0.79) |
| $p_{mask} = 0.50$ | 77.71 ($\pm$0.40) |
| $p_{mask} = 0.33$ | **78.15** ($\pm$0.23) |

Table 5: Performance comparison on *CommonsenseQA*'s development set, averaged across 5 training runs. $p_{mask}$ denotes the probability of masking an entity in the input sentence while training CoSe-Co; the masked entity is ensured to co-occur in the sentence and the target path.

Further, we obtain the optimal value of $p_{mask}$ by using the inferences obtained from the corresponding CoSe-Co configuration with the same supervised framework on a downstream task. Table 5 shows this for CSQA. We find that $p_{mask} = 0.33$ gives the best accuracy on the dev split followed by $p_{mask} = 0.5$, and thus we performed most of our ablations using the paths obtained from these two variants.

