# OpenReview forum: "CoSe-Co: Sentence Conditioned Generative CommonSense Contextualizer for Language Models"
_AKBC.ws/2021/Workshop/CSKB — CSKB_

### Official Review · Reviewer_8QEB · 2021-09-15
**CoSe-Co is used to generate contextually relevant commonsense inferences given a natural language input, recommend acceptance**

**Rating:** 8
**Confidence:** 5

**Review:**

Summary:
This paper focuses on generating contextually relevant commonsense inferences given a natural language input. The authors introduced a task agnostic sentence-conditioned generative CommonSense Contextualizer (CoSe-Co). CoSe-Co is observed to contain novel concepts that are relevant to the entire sentence context. The authors evaluate CoSe-Co on multi-choice QA and open-ended commonsense reasoning tasks on the CSQA, ARC, QASC, and OBQA datasets and demonstrate SOTA performance on these tasks. In addition, the authors also introduce a diverse-path search method to balance the quality and diversity of generated commonsense inferences.

Pros:
The paper is overall well-written and addresses an important/well-motivated question. The authors explain each step (Sentence-Path Paired Dataset Creation; Sentence to Commonsense Path Generator; Path Decoding During Inference) of the CoSe-Co pipeline very clearly. The authors adopt careful evaluation metrics concerning both relevance and diversity of generated concepts. The authors also show performance improvement across CSQA, ARC, QASC, and OBQA using the CoSe-Co they propose. I would recommend accepting the paper.

Questions:
Do the authors plan to release the dataset and the model publically?
For the CommonsenseQA experiments, the authors include the RoBERTa-large as a baseline. But since CoSe-Co is based on T5-base, it would be helpful to include T5-base results as well.
In Table 3, for the ARC task, it seems that the T5-base baseline is the best among all models, but T5-base+Concepts from CoSe-Co is labeled as the best one.

---

### Decision · Program_Chairs · 2021-09-18

Accept